# Autophagy-dependent ribosomal RNA degradation is essential for maintaining nucleotide homeostasis during *C. elegans* development

Yubing Liu[1,2,3], Wei Zou[2], Peiguo Yang[3], Li Wang[3,4], Yan Ma[2], Hong Zhang[3,4], Xiaochen Wang[3,4]*

[1]Peking University-Tsinghua University-National Institute of Biological Joint Graduate Program, School of Life Sciences, Peking University, Beijing, China; [2]National Institute of Biological Science, Beijing, China; [3]National Laboratory of Biomacromolecules, CAS Center for Excellence in Biomacromolecules, Institute of Biophysics, Chinese Academy of Sciences, Beijing, China; [4]College of Life Sciences, University of Chinese Academy of Sciences, Beijing, China

**Abstract** Ribosome degradation through the autophagy-lysosome pathway is crucial for cell survival during nutrient starvation, but whether it occurs under normal growth conditions and contributes to animal physiology remains unaddressed. In this study, we identified RNST-2, a *C. elegans* T2 family endoribonuclease, as the key enzyme that degrades ribosomal RNA in lysosomes. We found that loss of *rnst-2* causes accumulation of rRNA and ribosomal proteins in enlarged lysosomes and both phenotypes are suppressed by blocking autophagy, which indicates that RNST-2 mediates autophagic degradation of ribosomal RNA in lysosomes. *rnst-2(lf)* mutants are defective in embryonic and larval development and are short-lived. Remarkably, simultaneous loss of RNST-2 and de novo synthesis of pyrimidine nucleotides leads to complete embryonic lethality, which is suppressed by supplements of uridine or cytidine. Our study reveals an essential role of autophagy-dependent degradation of ribosomal RNA in maintaining nucleotide homeostasis during animal development.

DOI: https://doi.org/10.7554/eLife.36588.001

*For correspondence:
wangxiaochen@ibp.ac.cn

**Competing interests:** The authors declare that no competing interests exist.

## Introduction

Macroautophagy (hereafter referred to as autophagy) delivers cytoplasmic materials such as protein aggregates and organelles to lysosomes for degradation. The resulting catabolites are reutilized to maintain cellular homeostasis under both stress and physiological conditions (*Mizushima, 2009*; *Levine et al., 2011*). During autophagy, bulk cytosol may be randomly sequestered into double-membrane autophagosomes, which eventually fuse with lysosomes where cargos are degraded (*Xie and Klionsky, 2007*; *Nakatogawa et al., 2009*). On the other hand, specific substrates can be recognized and removed through selective autophagy (*Cebollero et al., 2012*). In both cases, multiple protein complexes act coordinately to control the initiation, formation and maturation of autophagosomes as well as their fusion with lysosomes (*Nakatogawa et al., 2009*; *Yang and Klionsky, 2010*).

A variety of cellular components including organelles are found to be substrates of autophagy. Ribosomes were among the first few cargos detected in the interior of autophagosomes by electron microscopy and have served as a marker of bulk cytoplasm degradation (*Ashford and Porter, 1962*; *Eskelinen et al., 2011*). In addition to non-selective autophagy, prolonged nitrogen starvation in

yeast leads to ribophagy, during which mature ribosomes are selectively targeted and removed by autophagy (*Kraft et al., 2008*). In ribophagy, the large and small ribosomal subunits appear to be independently targeted for degradation, which involves both ubiquitination and deubiquitination (*Kraft et al., 2008*; *Kraft and Peter, 2008*; *Ossareh-Nazari et al., 2010*; *Ossareh-Nazari et al., 2014*; *Dargemont and Ossareh-Nazari, 2012*). Ribosomes contain about 50% of the cellular proteins and 80% of total RNA. Both ribosomal proteins and RNAs are processed in lysosomes during autophagy-mediated degradation, which may serve as the major source of amino acids and nucleotides in nutrient deprivation conditions (*Huang et al., 2015*; *Lafontaine, 2010*). It is therefore not surprising that autophagic degradation of ribosomes is crucial for survival of yeast cells in nutrient starvation (*Kraft et al., 2008*). However, it remains unclear whether mature ribosomes are degraded through autophagy under normal growth conditions and whether ribosome degradation contributes to animal physiology.

*C. elegans* embryos are wrapped in tough eggshells that are impermeable to most solutes. The development of embryos from the one-cell stage to the end of embryogenesis (558 cells) relies on degradation of maternally loaded materials, independent of external nutrients. The maternally loaded yolk proteins are degraded in lysosomes, while PGL granules and aggregates of SQST-1/p62 and SEPA-1 family proteins are removed by autophagy (*Lin et al., 2013*; *Tian et al., 2010*; *Zhang et al., 2009*; *Liu et al., 2012*). The resulting catabolites are recycled from lysosomes to provide energy and essential building blocks for embryogenesis. Impairing autophagy or lysosome function leads to reduced hatching rate and retarded embryonic development (*Zhao et al., 2009*; *Tian et al., 2009*; *Liu et al., 2012*; *Sun et al., 2011*). It is unclear whether ribosomes, which contain abundant proteins and RNAs, may be used as a nutrient source for development.

In this study, we identified RNST-2, the *C. elegans* T2 family endoribonuclease, from a genetic screen for lysosome-defective mutants. We found that loss of RNST-2 causes accumulation of rRNA and ribosomal proteins in lysosomes in an autophagy-dependent manner. *rnst-2(lf)* worms are defective in embryonic and larval development and have shortened lifespans. Our data indicate that autophagy-dependent degradation of ribosomal RNA is important for maintaining nucleotide homeostasis, which is essential for development.

## Results

### *qx245* mutants contain abnormal lysosomes

From a forward genetic screen for mutants with lysosomal defects, we isolated *qx245*, which contained abnormally enlarged lysosomes. In wild type, lysosomes labeled by the lysosomal membrane protein LAAT-1::GFP and the lysosomal DNase II NUC-1::CHERRY appeared as small puncta and thin tubules, with an average volume of 0.77 $\mu m^3$ in 4-fold embryos and L1 larvae (*Figure 1A,D,G,J, M* and *Figure 1—figure supplement 1A,D,G,J,M,P*) (*Guo et al., 2010*; *Liu et al., 2012*). In *qx245* mutants, however, enlarged lysosomes were observed at embryonic, larval and adult stages in multiple cell types including hypodermal cells, muscle cells and sheath cells (*Figure 1B,E,H,K* and *Figure 1—figure supplement 1B,E,H,J,K,M,N*). The average volume of lysosomes reached 2.85 $\mu m^3$ in *qx245* embryos and L1 larvae, and 5.67 $\mu m^3$ in adult hypodermis, which is 3.7 and 2.5 times bigger than in wild type, respectively (*Figure 1M* and *Figure 1—figure supplement 1P*). The enlargement of lysosomes in *qx245* was observed from late embryonic to early larval stages, and at the adulthood (*Figure 1—figure supplement 1Q*). We found that the enlarged lysosomes in *qx245* mutants were stained by Lysotracker Red to a similar extent as in wild type, suggesting that lysosome acidification is not affected (*Figure 1J–K", N* and *Figure 1—figure supplement 1R*).

### *qx245* affects RNST-2, a widely expressed lysosomal T2 endoribonuclease

The *qx245* mutation affects the gene *rnst-2*, which encodes a T2 family endoribonuclease (RNase T2) (*Figure 1—figure supplement 2A and L*). RNase T2 family enzymes catalyze cleavage of single-stranded RNA at all four bases with an optimal pH of 4–5 (*Luhtala and Parker, 2010*). These enzymes perform diverse functions in a wide variety of organisms by processing extracellular and intracellular RNAs (*Luhtala and Parker, 2010*). We found that *qx245* carried a G-to-A mutation in *rnst-2* that resulted in the replacement of Gly 119 by Glu (*Figure 1—figure supplement 2L*). *bp555*,

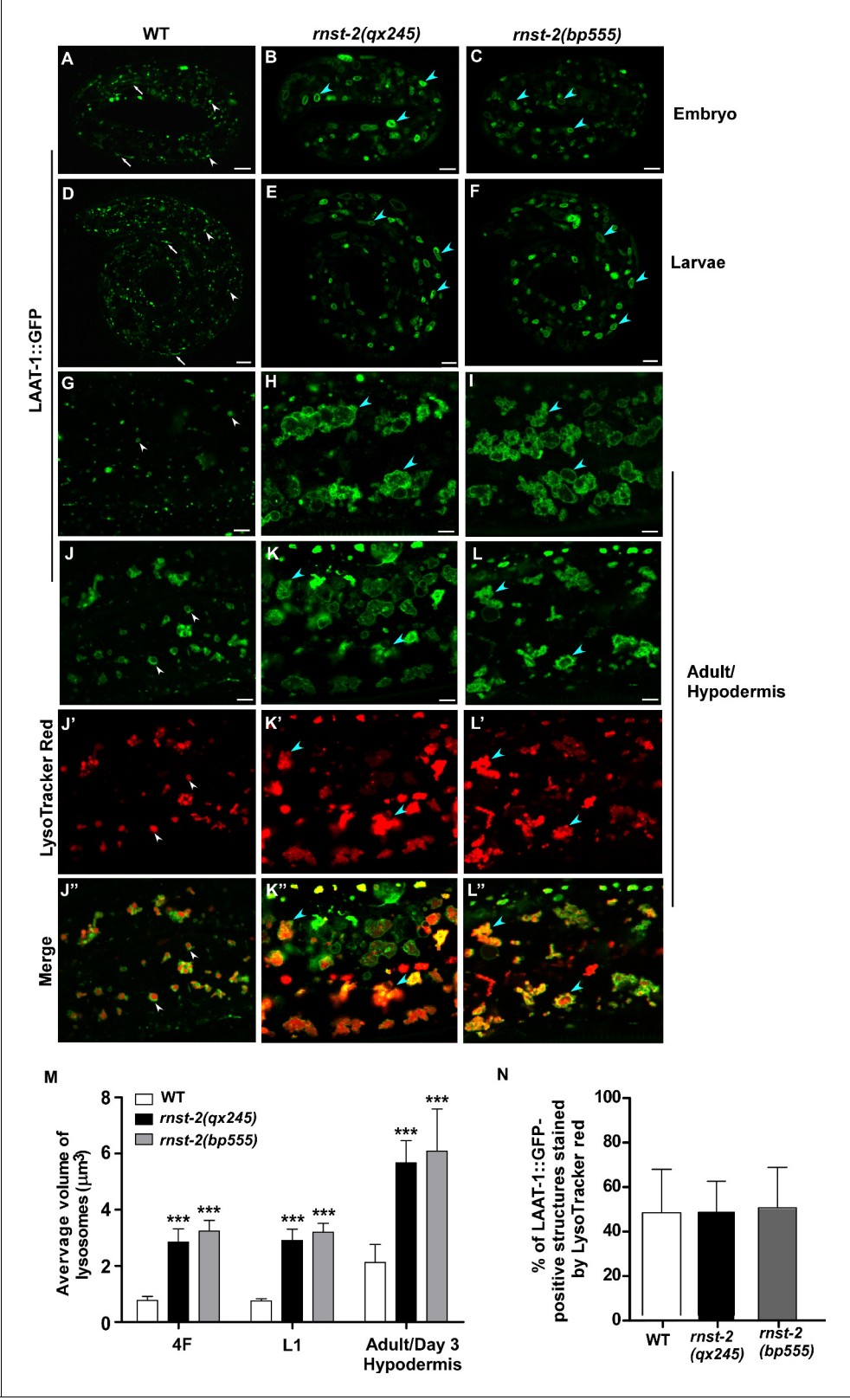

**Figure 1.** *rnst-2* mutants accumulate enlarged lysosomes. (A–I) Confocal fluorescence images of embryos at the 4-fold stage (4F, A–C), larvae 1 (L1, D–F) and adult hypodermis (G–I) in wild-type (WT, A, D, G), *rnst-2(qx245)* (B, E, H) and *rnst-2(bp555)* (C, F, I) expressing LAAT-1::GFP. (J–L") Confocal fluorescence images of the hypodermis in wild-type (J–J"), *rnst-2 (qx245)* (K–K") and *rnst-2(bp555)* (L–L") adults expressing LAAT-1::GFP and stained by Lysotracker red. In (A–L''), white arrowheads and arrows indicate globular and tubular lysosomes, respectively, and blue arrowheads indicate enlarged

*Figure 1 continued on next page*

*Figure 1 continued*

globular lysosomes. Scale bars: 5 µm. (**M**) Quantification of the average volume of lysosomes labeled by LAAT-1::GFP in 4-fold-stage embryos (4F), L1 larvae (L1) and adult hypodermis (day 3 of adulthood). (**N**) The percentage of LAAT-::GFP-positive lysosomes that were stained by lysotracker red was quantified in adult hypodermis. In (**M, N**), at least 10 worms were scored in each strain at each stage. Data are shown as mean ± SD. Two-way ANOVA with the Bonferroni *post hoc* test (**M**) or one-way ANOVA with Tukey's *post hoc* test (**N**) was performed to compare mutant datasets with wild type. ***$p<0.001$, other points had $p>0.05$.

DOI: https://doi.org/10.7554/eLife.36588.002

The following source data and figure supplements are available for figure 1:

**Source data 1.** *Figure 1* numerical data.
DOI: https://doi.org/10.7554/eLife.36588.005
**Source data 2.** *Figure 1—figure supplement 1* numerical data.
DOI: https://doi.org/10.7554/eLife.36588.006
**Figure supplement 1.** *rnst-2* mutants accumulate enlarged lysosomes in multiple cell types.
DOI: https://doi.org/10.7554/eLife.36588.003
**Figure supplement 2.** Molecular cloning of *rnst-2*.
DOI: https://doi.org/10.7554/eLife.36588.004

an independently isolated *rnst-2* mutant allele, caused a premature stop codon after Pro 55. The G119E mutation in *qx245* affected the second catalytic active site (CAS II), whereas both CAS I and II were lost in *bp555*, suggesting that *qx245* and *bp555* are strong loss-of-function or null mutations of *rnst-2* (*Figure 1—figure supplement 2L*). Consistent with this, enlarged lysosomes were observed in *bp555*, like in *qx245* (*Figure 1C,F,I,L,M* and *Figure 1—figure supplement 1C,F,I,L,O,P*). *C. elegans* RNST-2 shares 47% sequence similarity and 29% sequence identity with human RNASET2 (*Figure 1—figure supplement 2L*). Expression of RNST-2 or human RNASET2 controlled by the *rnst-2* promoter efficiently rescued the enlarged lysosome phenotype in *qx245* mutants, suggesting that RNASET2 can substitute for worm RNST-2 in maintaining lysosome morphology (*Figure 1—figure supplement 2C,E–G,I*).

We generated a RNST-2::CHERRY reporter driven by the *rnst-2* promoter, which fully rescued the lysosome phenotypes in *qx245* and *bp555* mutants (*Figure 1—figure supplement 2C,F,G,J,K*). RNST-2::CHERRY was widely expressed from embryonic stages throughout larval and adult stages in various tissues, including pharynx, hypodermis, muscle, sheath cells, intestine cells, vulva and tail region (*Figure 2A–H'*). RNST-2::CHERRY stained both puncta and tubular structures that were labeled by the lysosomal membrane proteins LAAT-1::GFP and SCAV-3::GFP, indicating that RNST-2 localizes to lysosomes (*Figure 2I–J''*).

## *rnst-2(lf)* lysosomes accumulate rRNA and ribosomal proteins in an autophagy-dependent manner

To determine whether RNST-2 functions as a ribonuclease in lysosomes, we performed rescue experiments. Expression of wild-type RNST-2, but not RNST-2(H118A), which carries a mutation in the histidine residue (H118) essential for the catalytic activity of endoribonuclease T2 (*Kawata et al., 1990*), rescued the enlarged lysosome phenotype of *rnst-2(qx245)* mutants, indicating that catalytic activity of RNST-2 is important for its function in lysosomes (*Figure 1—figure supplement 2C,D,G, H*). We purified lysosomes from wild type and *rnst-2* worms and extracted RNA from lysosomes (*Figure 3—figure supplement 1A,B*) (*Liu et al., 2012*). We found that *rnst-2(qx245)* and *rnst-2(bp555)* lysosomes accumulated high levels of RNA, especially 26S and 18S rRNA (*Figure 3A* and *Figure 3—figure supplement 1C*). The total RNA level in *rnst-2* lysosomes was more than five times higher than in wild type (*Figure 3A* and *Figure 3—figure supplement 1C*). Moreover, lysosomal accumulation of ribosomal proteins from the large and small subunits was observed in *rnst-2(qx245)* and *rnst-2(bp555)* but not wild-type worms (*Figure 3B* and *Figure 3—figure supplement 1D*). The lysosomal accumulation of RNA and ribosomal proteins in *rnst-2(qx245)* mutants was fully suppressed by expression of RNST-2 (*Figure 3A and B*). These data indicate that loss of RNST-2 disrupts ribosomal RNA degradation in lysosomes. The defective degradation of rRNA in *rnst-2* lysosomes may impair further digestion of ribosomes, causing accumulation of ribosomal proteins in lysosomes.

We next investigated whether autophagy is responsible for delivering ribosomal RNA to lysosomes. ATG-2/ATG2 and EPG-6/WIPI4 regulate autophagosome formation and loss of their

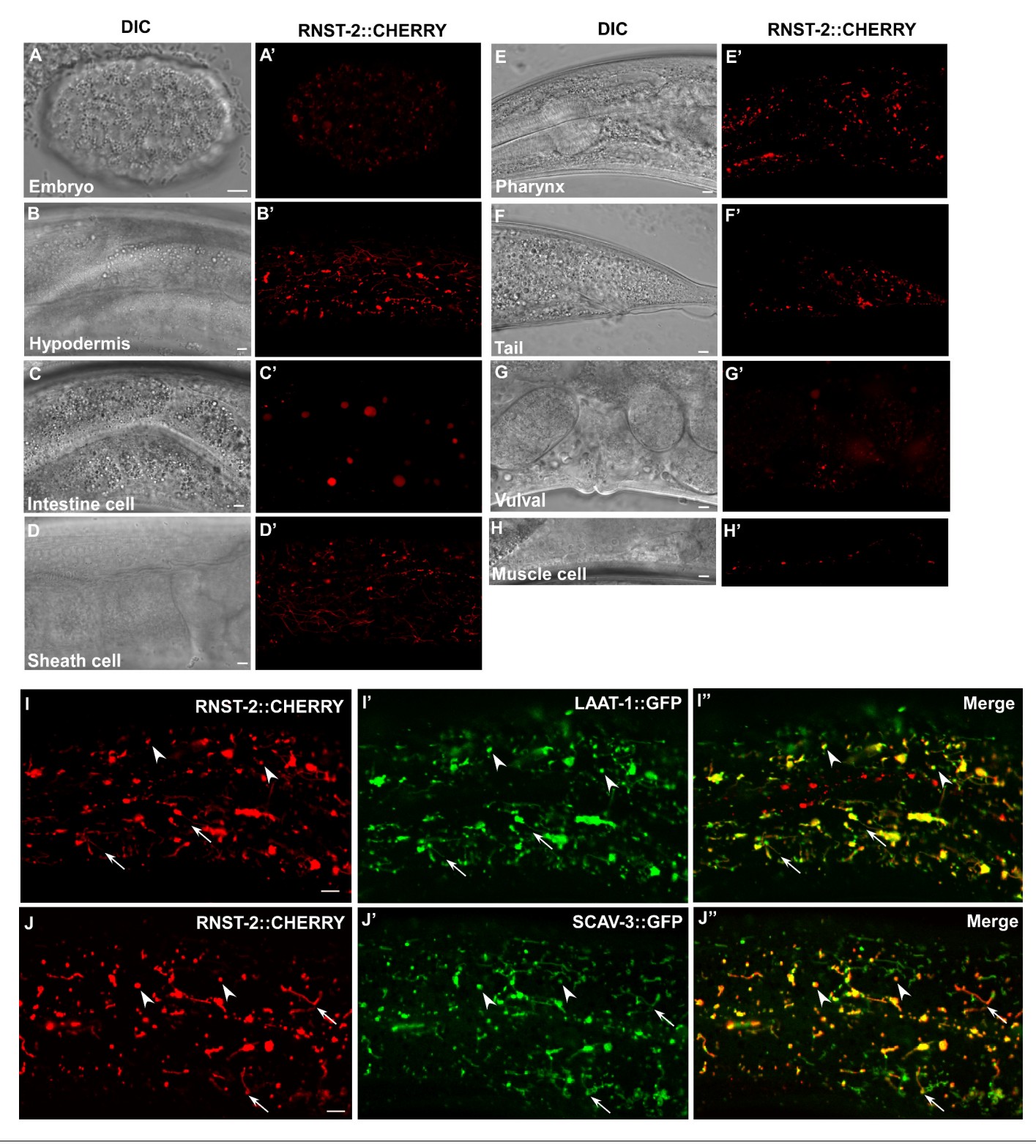

**Figure 2.** RNST-2 is widely expressed and localizes to lysosomes. (**A–H'**) DIC and confocal fluorescence images of wild type expressing RNST-2::CHERRY driven by the *rnst-2* promoter. RNST-2::CHERRY is expressed from early embryos (**A, A'**) to the adult stage in various cell types including hypodermis (**B, B'**), intestine (**C, C'**), sheath cell (**D, D'**), pharynx (**E, E'**), tail (**F, F'**), vulva (**G, G'**) and muscle cell (**H, H'**). (**I–J''**) Confocal fluorescence images of the hypodermis in wild type co-expressing RNST-2::CHERRY and LAAT-1::GFP (**I–I''**) or SCAV-3::GFP (**J–J''**). RNST-2 colocalizes with LAAT-1 and SCAV-3 to both globular (arrowheads) and tubular (arrows) lysosomes. Scale bars: 5 µm.
DOI: https://doi.org/10.7554/eLife.36588.007

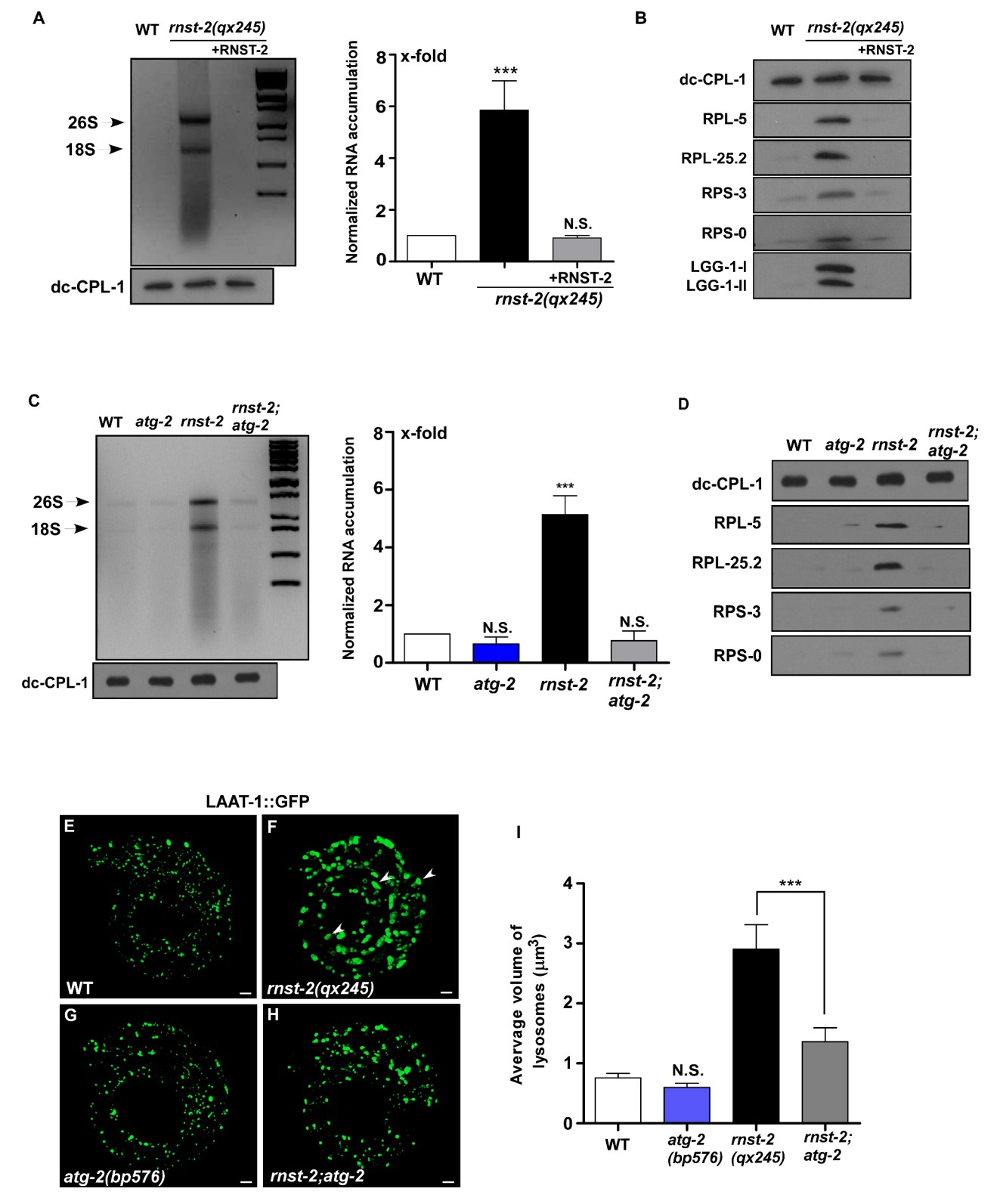

**Figure 3.** Loss of RNST-2 causes accumulation of rRNA and ribosomal proteins in lysosomes in an autophagy-dependent manner. (**A, C**) RNA purified from lysosomes in the indicated strains was examined by agarose gel electrophoresis. Full processing of the lysosomal cathepsin CPL-1 (dc-CPL-1) was used to normalize the amount of lysosomes in each strain. The total extracted RNA was quantified and normalized to 1-fold in wild type (A, C, right panel). Abundant 26S and 18S rRNA was observed in lysosomes of *rnst-2(qx245)* mutants. At least three independent experiments were performed and *Figure 3 continued on next page*

*Figure 3 continued*

data are shown as mean ± SD. (B, D) Accumulation of ribosomal proteins (large subunit: RPL-5, RPL-25.2; small subunit: RPS-3 and RPS-0) and LGG-1 in lysosomes was examined by western blot analysis in the indicated strains. Full processing of the lysosomal cathepsin CPL-1 (dc-CPL-1) was used to normalize the amount of lysosomes in each strain. (E–H) 3D reconstitution of the fluorescence images in 10–15 z-series (0.5 μm/section) in L1 larvae of the indicated strains expressing LAAT-1::GFP. Enlarged lysosomes (arrowheads) were observed in *rnst-2(qx245)*. (I) Quantification of the average volume of lysosomes in the strains shown in (E–H). At least 10 worms were scored in each strain and data are shown as mean ± SD. In (A, C, I), one-way ANOVA with Tukey's *post hoc* test was performed to compare all other datasets with wild type or datasets that are linked by lines (I). ***p<0.0001, N.S.: no significance.

DOI: https://doi.org/10.7554/eLife.36588.008

The following source data and figure supplements are available for figure 3:

**Source data 1.** *Figure 3* numerical data.
DOI: https://doi.org/10.7554/eLife.36588.011
**Source data 2.** *Figure 3—figure supplement 1* numerical data.
DOI: https://doi.org/10.7554/eLife.36588.012
**Source data 3.** *Figure 3—figure supplement 2* numerical data.
DOI: https://doi.org/10.7554/eLife.36588.013
**Figure supplement 1.** Lysosomal accumulation of rRNA and ribosomal proteins in *rnst-2* is suppressed by blocking autophagy.
DOI: https://doi.org/10.7554/eLife.36588.009
**Figure supplement 2.** Loss of RNST-2 does not affect degradation of endocytic and phagocytic cargos.
DOI: https://doi.org/10.7554/eLife.36588.010

functions disrupt progression from phagophores to complete autophagosomes (*Tian et al., 2010*; *Lu et al., 2011*). We found that mutation of *atg-2* or *epg-6* completely suppressed lysosomal accumulation of rRNA and ribosomal proteins in *rnst-2(qx245)* mutants (*Figure 3C,D* and *Figure 3—figure supplement 1E,F*). Moreover, lysosome volume, which was greatly increased in *rnst-2(qx245)*, was reduced significantly in *rnst-2;atg-2* double mutants (*Figure 3E–I*). Loss of EPG-6 or LGG-1, the *C. elegans* homolog of Atg8/LC3, also led to greatly reduced lysosome volume in *rnst-2* (*Figure 3—figure supplement 1G–M*). Altogether, these data suggest that rRNA and ribosomal proteins are delivered to lysosomes through autophagy and defects in their degradation lead to enlargement of lysosomes in *rnst-2* mutants.

## Loss of RNST-2 does not affect endocytic and phagocytic cargo degradation but partially impairs autophagy

*rnst-2* mutants accumulate ribosomal RNA and proteins that are delivered to lysosomes by autophagy. We next examined whether loss of RNST-2 also affects degradation of endocytic and phagocytic cargos. Cell surface protein CAV-1 and yolk lipoprotein VIT-2 are delivered to lysosomes through the endocytic pathway and are degraded shortly after fertilization and during embryogenesis, respectively (*Sato et al., 2006*; *Grant and Hirsh, 1999*). In *rnst-2(qx245)* embryos, both VIT-2::GFP and CAV-1::GFP were properly degraded as in wild type (*Figure 3—figure supplement 2A–H'*). Moreover, *rnst-2(qx245)* mutants contained similar numbers of embryonic and germ cell corpses as in wild type, suggesting that phagocytosis and degradation of apoptotic cells is unaffected (*Figure 3—figure supplement 2I,J*). These data suggest that lysosomal degradation of endocytic and phagocytic cargos is not affected in *rnst-2(lf)* mutants. *rnst-2(lf)* lysosomes contained high levels of LGG-1, which associates with autophagosomes and their precursors (*Figure 3B*). This lysosomal accumulation of LGG-1 is consistent with defects in degrading autophagic cargos (*Tian et al., 2010*). We next examined whether the autophagy process is affected in *rnst-2* mutants. The PGL granule component PGL-3 is removed in soma by selective autophagy during embryogenesis, which requires the bridging molecule SEPA-1 (*Zhang et al., 2009*). PGL-3 and SEPA-1 puncta persisted in autophagy-defective mutants such as *atg-2* (*Figure 4—figure supplement 1C,C', F,F'*) (*Zhang et al., 2009*), but were removed in both wild-type and *rnst-2* embryos, suggesting that autophagic clearance of PGL granules is unaffected in *rnst-2* mutants (*Figure 4—figure supplement 1A–B', D–E'*). SQST-1 is the *C. elegans* p62 homolog that associates with various protein aggregates and is removed by autophagy (*Tian et al., 2010*). SQST-1 puncta were cleared in wild-type embryos, present in late-staged *rnst-2* mutant embryos, and persisted at both early and late stages in *atg-2* embryos (*Figure 4A–I'* and *Figure 4—figure supplement 1G–I'*). Similarly, LGG-1, which associates

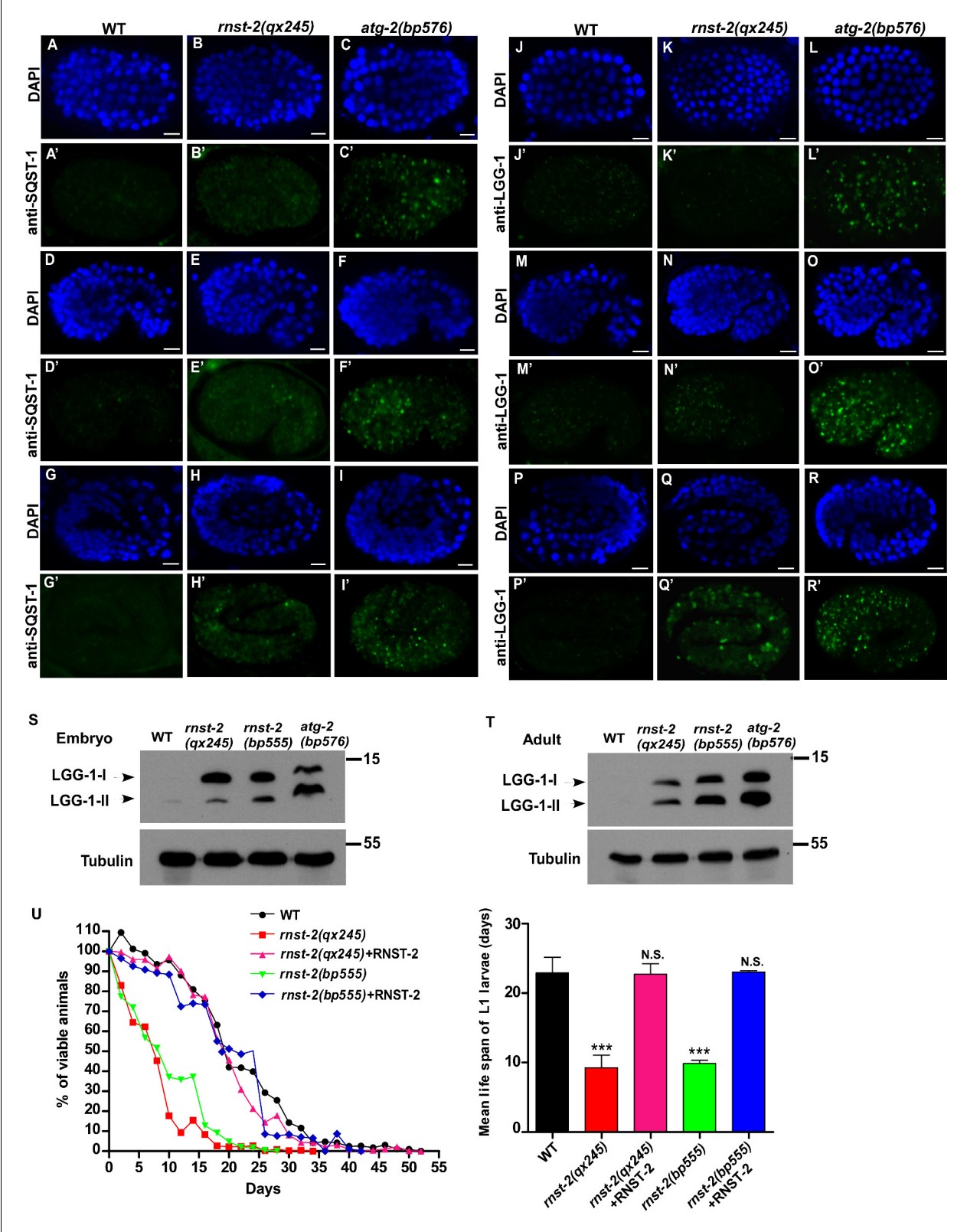

**Figure 4.** Autophagy is partially impaired in *rnst-2* mutants. (A–R') Confocal fluorescence images of embryos in wild type (WT), *rnst-2(qx245)* and *atg-2 (bp576)* at 200 cell (A–C', J–L"), comma (D–F', M–O') and 4-fold (G–I', P–R') stages stained by anti-SQST-1 (A–I') or anti-LGG-1 antibodies (J–R'). DAPI staining shows nuclei in each embryo. Scale bars: 5 μm. (S, T) Western blot analysis of LGG-1-I and LGG-1-II (lipid-conjugated form) in wild-type, *rnst-2 (qx245)*, *rnst-2(bp555)* and *atg-2(bp576)* at the embryonic (S) and adult stages (day 3 of adulthood) (T). (U) The survival of L1 larvae in the absence of

*Figure 4 continued on next page*

*Figure 4 continued*
food was quantified in the indicated strains. At least 200 animals were scored at each time point in each strain. three independent experiments were performed and the mean lifespan of L1 larvae in the absence of food was quantified (right panel). The data are shown as mean ± SD. One-way ANOVA with Tukey's *post hoc* test was performed to compare mutant datasets with wild type. ***p<0.0001, N.S., no significance.
DOI: https://doi.org/10.7554/eLife.36588.014
The following source data and figure supplements are available for figure 4:

**Source data 1.** *Figure 4* numerical data.
DOI: https://doi.org/10.7554/eLife.36588.017
**Figure supplement 1.** Autophagic clearance of PGL granule is not affected in *rnst-2* mutants.
DOI: https://doi.org/10.7554/eLife.36588.015
**Figure supplement 2.** GFP::ATG-18 is diffuse in the cytoplasm of wild-type and *rnst-2* mutant embryos.
DOI: https://doi.org/10.7554/eLife.36588.016

with autophagic structures and is a substrate of autophagy, was present in *atg-2* embryos at both early and late stages, but accumulated only in late-staged embryos in *rnst-2* mutants (*Figure 4J–R'* and *Figure 4—figure supplement 1J–L'*). In addition, GFP::ATG-18, the PtdIns3P-binding protein that associates with early autophagic structures (*Lu et al., 2011*), was diffuse in the cytoplasm in both wild type and *rnst-2(lf)*, suggesting that autophagosome formation is not blocked (*Figure 4—figure supplement 2*). Altogether, these data suggest that the autophagy process is partially impaired or delayed in *rnst-2(lf)*, causing accumulation of SQST-1 and LGG-1 in late-staged embryos. Consistent with this, *rnst-2* mutants accumulated both LGG-1-I and LGG-1-II (lipid-conjugated form of LGG-1) at embryonic and adult stages (*Figure 4S and T*). Moreover, loss of *rnst-2* significantly shortened the lifespan of the L1 larvae in the absence of food (*Figure 4U*), a process that requires autophagy activity (*Kang et al., 2007*).

## *rnst-2* mutants are defective in embryogenesis and larval development, and are short-lived

*rnst-2* mutants are viable but grow slowly. Both *rnst-2(qx245)* and *rnst-2(bp555)* were retarded in embryogenesis and exhibited 20–30% embryonic lethality (*Figure 5A and B*). Moreover, approximately 50% of *rnst-2(qx245)* and *rnst-2(bp555)* embryos that hatched were arrested during larval development, and this arrest was rescued by expression of RNST-2 (*Figure 5C*). These data indicate that loss of *rnst-2* severely affects embryonic and larval development. In addition to affecting development, *rnst-2* mutants are short-lived compared with wild type (*Figure 5D and E*). To further investigate the effect of *rnst-2(lf)* on lifespan, we examined whether loss of *rnst-2* affects the lifespan of long-lived mutants. Reducing insulin signaling extends the lifespan in *C. elegans* (*Murphy and Hu, 2013*). Worms that carry a loss-of-function mutation in the insulin receptor DAF-2 live twice as long as wild-type worms (*Figure 5D and E*) (*Kenyon et al., 1993*). We found that loss of *rnst-2* significantly reduced the lifespan of *daf-2(lf)* worms to the wild-type level (*Figure 5D and E*). Moreover, loss of *rnst-2* also reduced the lifespan of *glp-1(e2144)*, which affects germline stem cells and thus extends lifespan (*Figure 5F and G*) (*Arantes-Oliveira et al., 2002*). These data suggest that RNST-2 function is important for maintaining normal lifespan and for the lifespan extension in *daf-2* and *glp-1*.

## Simultaneous loss of *rnst-2* and de novo synthesis of pyrimidines leads to synthetic embryonic lethality

RNST-2 is an endoribonuclease that degrades RNA in lysosomes. We suspected that defective RNA degradation in *rnst-2(lf)* mutants affects generation and recycling of RNA catabolites, which are important for animal development. Pyrimidine and purine nucleotides are essential components of RNA. They are produced through both de novo and salvage pathways and are building blocks for the synthesis of RNA, DNA, phospholipids and nucleotide sugars (*Huang and Graves, 2003*). As purine metabolic pathways are not well understood in *C. elegans* and the key enzyme involved in purine biosynthesis is not clearly identified, we focused on the pyrimidine pathway. PYR-1/CAD and UMPS-1/UMPS are two key enzymes required for the de novo synthesis of UMP, the precursor for other pyrimidine nucleotides (*Merry et al., 2014*; *Franks et al., 2006*; *Huang and Graves, 2003*). *pyr-1(cu8)* and *umps-1(mn160)* single mutants exhibited 50% and 25% embryonic lethality,

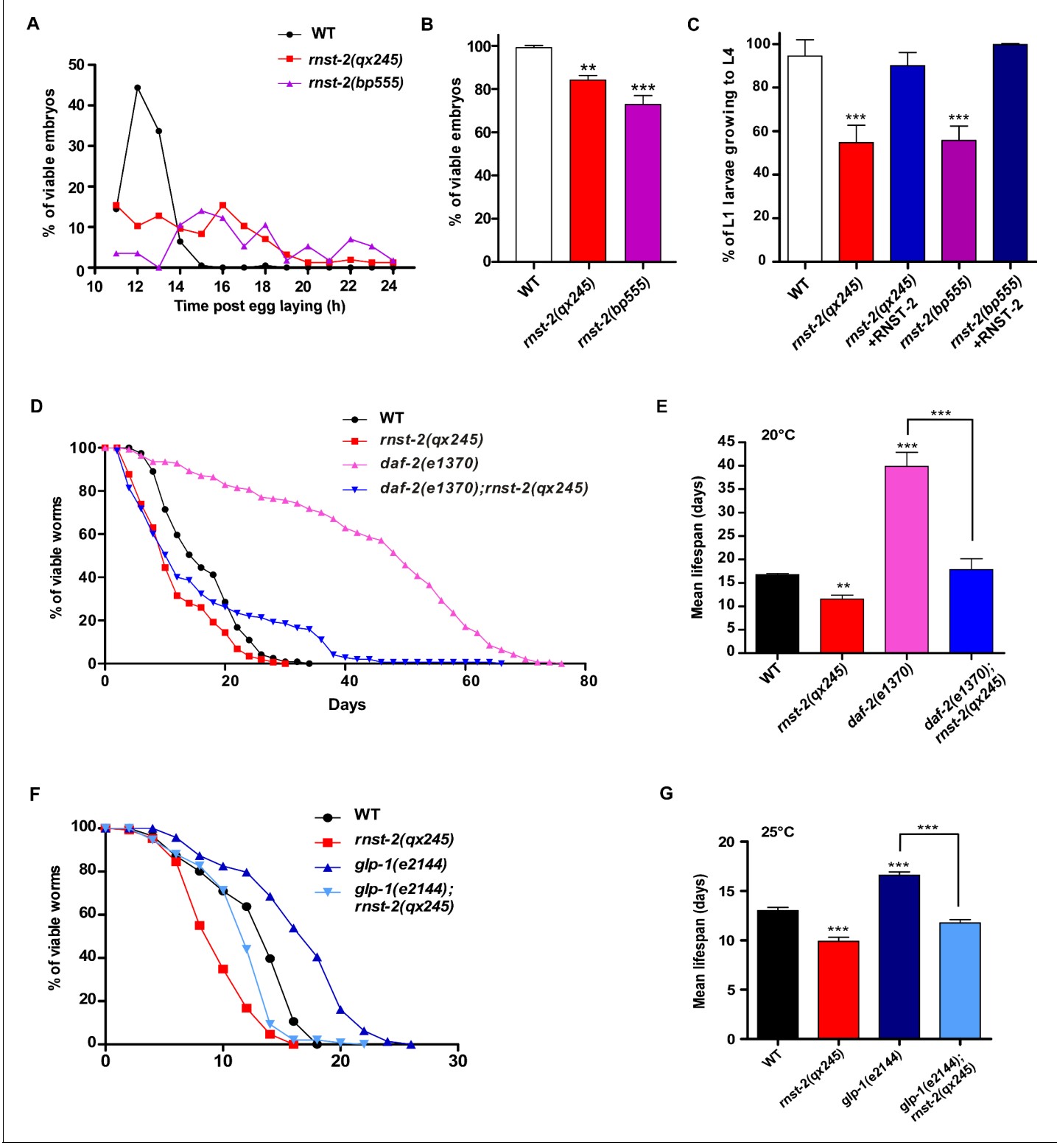

**Figure 5.** *rnst-2* mutants are defective in embryogenesis and larval development, and are short-lived. (**A–C**) Embryonic and larval development was examined in wild type, *rnst-2(qx245)* and *rnst-2(bp555)*. At least 150 embryos and 100 larvae were examined in each strain and at least three independent experiments were performed. (**D–G**) Lifespan analyses were performed in the indicated strains. More than 100 worms were examined in each strain and three independent experiments were performed. The mean lifespan in the indicated strains was quantified and is shown in (**E, G**). Data are shown as mean ± SD. One-way ANOVA with Tukey's *post hoc* test was performed to compare datasets with wild type or datasets that are linked by lines. *p<0.05, **p<0.001, ***p<0.0001.

*Figure 5 continued on next page*

*Figure 5 continued*

DOI: https://doi.org/10.7554/eLife.36588.018

The following source data is available for figure 5:

**Source data 1.** *Figure 5* numerical data.
DOI: https://doi.org/10.7554/eLife.36588.019

respectively, indicating that pyrimidine biosynthesis is important for embryogenesis (*Figure 6A and B*). We found that loss of *rnst-2* and *pyr-1* or *rnst-2* and *umps-1* led to almost 100% embryonic lethality (*Figure 6A,B* and *Figure 6—figure supplement 1A,B*). By contrast, loss of the lysosomal amino acid transporter LAAT-1, which blocks transport of lysine and arginine out of lysosomes, did not cause increased embryonic lethality in *pyr-1* or *umps-1* mutants (*Figure 6A and B*) (*Liu et al., 2012*). Moreover, loss of LMP-1, the lysosomal membrane protein homologous to human LAMP1, did not affect embryonic and larval development, and had no effect on the embryonic viability in *pyr-1* or *umps-1* (*Figure 6—figure supplement 1C–E*). These data suggest that loss of RNST-2 function, but not a general defect of lysosomes, leads to the synthetic embryonic lethality in *pyr-1* or *umps-1* mutants. Similar to *rnst-2(lf)*, the *pyr-1(cu8)* mutation also caused a significantly shortened lifespan in *daf-2(lf)* worms, whereas *laat-1(qx42)* was less effective in reducing the *daf-2(lf)* lifespan (*Figure 6—figure supplement 1F–I*). These data suggest that maintenance of pyrimidine homeostasis is important for extended lifespan in *daf-2(lf)*.

To further test whether lysosomal degradation of rRNA by RNST-2 is important in providing pyrimidines during embryonic development, we performed pyrimidine supplement experiments. External supplement of uridine had no effect on embryogenesis in wild type, but efficiently suppressed the synthetic embryonic lethality in *pyr-1;rnst-2* double mutants (*Figure 6C*). The embryonic lethality in *umps-1;rnst-2* was partially relieved by uridine supplement, consistent with the idea that UMPS-1 acts in both de novo and salvage pathways to synthesize UMP (*Figure 6D*) (*Merry et al., 2014*). We found that cytidine supplement caused reduced viability of wild-type embryos, but it partially suppressed the synthetic embryonic lethality in *pyr-1;rnst-2* and *umps-1;rnst-2* double mutants, albeit to a lesser extent than uridine supplements (*Figure 6E and F*). These data suggest that RNA degradation by RNST-2 in lysosomes is important in maintaining pyrimidine homeostasis during embryogenesis. We found that the autophagy-defective mutations *lgg-1(bp500)*, *atg-2(bp576)*, *atg-18(gk378)* and *atg-9(bp564)* partially affected embryogenesis and caused synthetic embryonic lethality in *pyr-1 (RNAi)* and *umps-1(RNAi)* worms, consistent with the role of autophagy in delivering RNA to lysosomes (*Figure 6—figure supplement 1A,B*). Supplements of uridine or cytidine did not obviously improve the embryonic viability of *rnst-2(qx245)* single mutants or in autophagy-defective mutants (*Figure 6C–F* and *Figure 6—figure supplement 1C,D*), suggesting that a general autophagy defect may contribute to the partial embryonic lethality in these worms.

## Discussion

### RNST-2 degrades ribosomal RNA delivered by autophagy in lysosomes

Here we identified *C. elegans* RNST-2, a T2 family endoribonuclease, as a key enzyme that degrades ribosomal RNA in lysosomes. RNST-2 localizes to lysosomes and loss of its function causes rRNA accumulation in enlarged lysosomes in an autophagy-dependent manner. Thus, RNST-2 degrades rRNA delivered by autophagy in lysosomes, which is consistent with the function of Ryn1 and RNS2, the yeast and *Arabidopsis thaliana* T2 family endoribonucleases, respectively (*Huang et al., 2015*; *Floyd et al., 2015*). Loss of zebrafish *rnaset2* results in accumulation of undigested rRNA in lysosomes of neurons (*Haud et al., 2011*), while expression of human RNASET2 in worms efficiently rescued the lysosome phenotype in *rnst-2* mutants (*Figure 1—figure supplement 2E,I*). Thus, T2 family endoribonuclease acts as the key enzyme to degrade ribosomal RNA in lysosomes in diverse species throughout evolution. The *rnst-2* lysosomes appear to accumulate intact 26S and 18S rRNA, suggesting that mature ribosomes are targeted for degradation. In addition to rRNA, *rnst-2* lysosomes also contain high levels of ribosomal proteins, suggesting that rRNA degradation may facilitate destruction and further digestion of ribosomes. Our data suggest that the autophagy process is partially

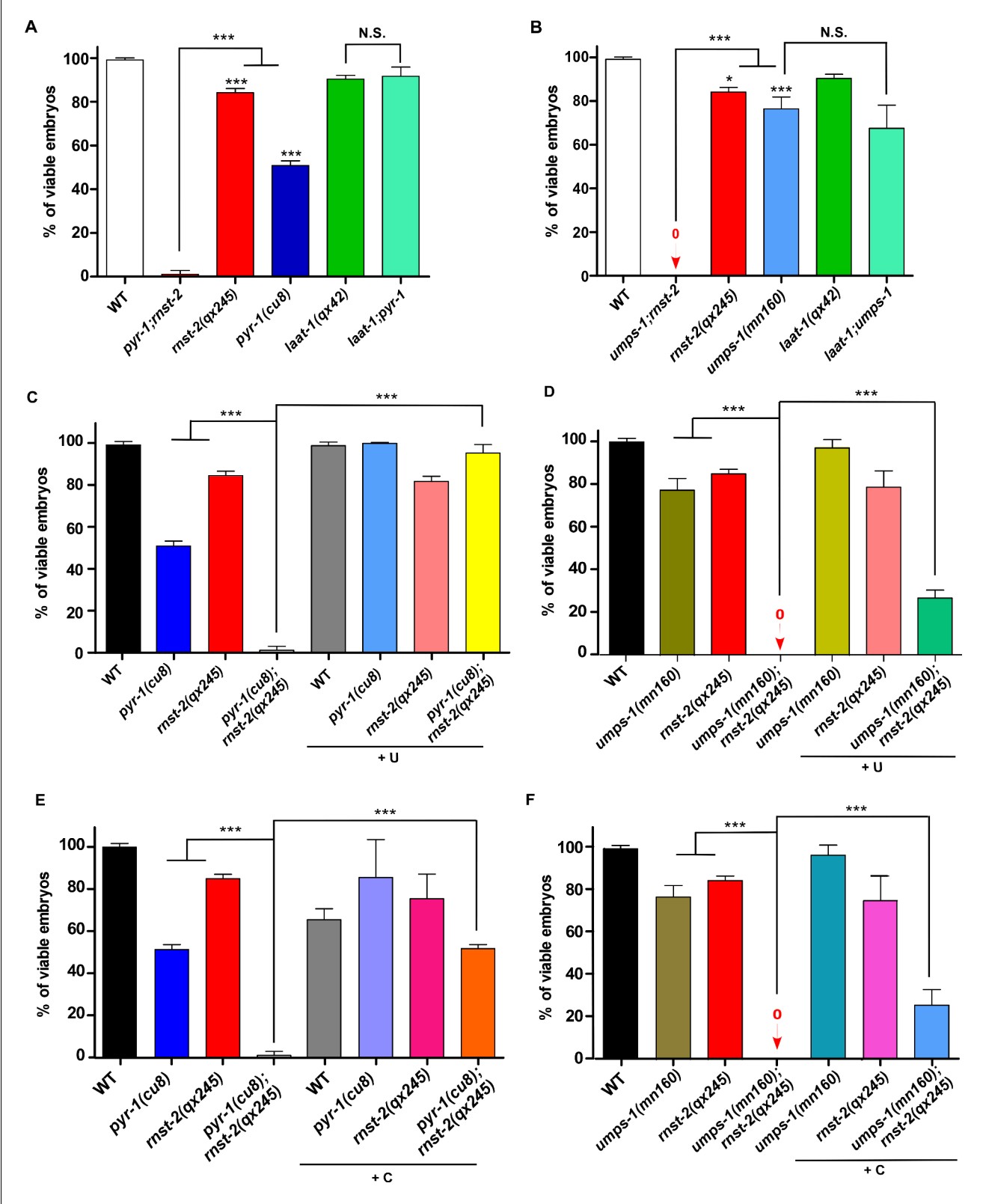

**Figure 6.** RNST-2 maintains pyrimidine availability for embryonic development. The percentage of viable embryos was scored in the indicated strains without (**A, B**) or with uridine (**C, D**) or cytidine supplementation (**E, F**). At least 150 embryos were examined in each strain and three independent experiments were performed. Data are shown as mean ± SD. One-way ANOVA with Tukey's *post hoc* test was performed to compare all other datasets with wild type or datasets that are linked by lines. *p<0.05, **p<0.001, ***p<0.0001, N.S.: no significance.

*Figure 6 continued on next page*

*Figure 6 continued*

DOI: https://doi.org/10.7554/eLife.36588.020

The following source data and figure supplements are available for figure 6:

**Source data 1.** *Figure 6* numerical data.
DOI: https://doi.org/10.7554/eLife.36588.023
**Source data 2.** *Figure 6—figure supplement 1* numerical data.
DOI: https://doi.org/10.7554/eLife.36588.024
**Source data 3.** *Figure 6—figure supplement 2* numerical data.
DOI: https://doi.org/10.7554/eLife.36588.025
**Figure supplement 1.** Loss of RNST-2 function but not a general lysosomal defect leads to synthetic embryonic lethality in *pyr-1* and *umps-1* mutants.
DOI: https://doi.org/10.7554/eLife.36588.021
**Figure supplement 2.** Autophagy is important for maintaining pyrimidine availability during embryonic development.
DOI: https://doi.org/10.7554/eLife.36588.022

impaired in *rnst-2* mutants. We suspect that gradual buildup of undigested rRNA and ribosomal proteins in lysosomes leads to impaired autophagy kinetics and thus reduction in autophagic flux. Consistent with this, dysregulation of autophagy has been commonly found in lysosomal storage disorders characterized by the accumulation of undegraded metabolites in lysosomes (*Seranova et al., 2017*).

## Ribosomal RNA degradation through the autophagy-lysosome pathway is important in maintaining nucleotide homeostasis essential for animal development

Bulk or selective degradation of ribosomes via the autophagy-lysosome pathway is important for cell survival under nutrient deprivation conditions, but whether it occurs under normal conditions and is important for animal physiology remains unclear. Here, we found that loss of RNST-2 function affects degradation of ribosomal RNA and proteins, and causes severe developmental defects, suggesting that ribosome degradation occurs under normal growth conditions and is important for animal development. Importantly, simultaneous loss of RNST-2 or autophagy and the de novo synthesis of pyrimidines causes complete embryonic lethality, and this synthetic lethality can be suppressed by exogenous supply of uridine or cytidine. This indicates that rRNA turnover through the autophagy-lysosome pathway plays an important role in maintaining nucleotide homeostasis during development. We have attempted to measure the nucleoside levels by LC/MS in *rnst-2;pyr-1(RNAi)* embryos, and observed a reduction in cytidine but not guanosine (data not shown). Given that the nucleoside levels were measured in embryonic lysates prepared from unhealthy embryos with an intact purine synthesis pathway but containing heterogeneous tissues and both intracellular and extracellular nucleosides, the results may not reflect the situation in key tissues that cause the phenotype.

The nucleosides or nucleobases derived from lysosomal rRNA turnover may be re-utilized through the pyrimidine salvage pathway to provide precursors to multiple metabolic pathways essential for development (*Huang et al., 2015*; *Huang and Graves, 2003*). In *rnst-2* embryos, lysosome enlargement was not observed at the early phase of embryogenesis (before comma) when cell proliferation and organogenesis occur, but was very obvious after the 2-fold stage, during which extensive morphogenesis occurs and cell proliferation almost ceases (*Figure 1—figure supplement 1Q*) (*Riddle et al., 1997*). It is possible that excessive ribosomes are removed at the late phase of embryogenesis to cope with the lower demand for protein synthesis and the high levels of energy that are required for morphogenesis and hatching. Although supplements of uridine or cytidine significantly reduced the embryonic lethality in *rnst-2;pyr-1* double mutants, it did not obviously improve the embryonic viability of *rnst-2(qx245)* single mutants, suggesting that the developmental phenotype may be attributed to defects in both general autophagy and nucleotide supply.

In addition to late embryonic and early larval stages, enlarged lysosomes are also present in *rnst-2* adults, and the *rnst-2* mutation reduces the lifespan of both wild-type and long-lived *daf-2* and *glp-1* worms. This suggests that lysosomal degradation of ribosomes also occurs during adulthood and may contribute to longevity. Lysosome-dependent ribosome turnover may help to clear defective ribosomes generated during aging and/or contribute to downregulation of protein translation. It is reported recently that NUFIP1 serves as a receptor for the selective autophagy of ribosomes in

mammalian cells under nutrient starvation conditions to supply nucleosides required for cell survival during starvation (*Wyant et al., 2018*). Here we found that ribosome degradation through the autophagy-lysosome pathway occurs under normal growth conditions and is important for development and longevity. Whether ribosomes are recognized and removed through selective or bulk autophagy during development and aging requires future investigations.

Our study reveals the essential role of autophagy-dependent ribosome degradation in animal development, which may be conserved through evolution as both the autophagy machinery and T2 family endoribonucleases are highly similar from worms to humans. Loss-of-function mutations in human RNASET2 lead to familial cystic leukoencephalopathy, which is considered as a lysosome storage disease, but how lysosomal rRNA accumulation leads to neuronal disease is unclear (*Henneke et al., 2009*; *Haud et al., 2011*). Our finding that loss of *rnst-2* affects nucleotide homeostasis and partially impairs autophagy may provide further insights into the disease pathogenesis.

# Materials and methods

## Key resources table

| Reagent type (species) or resource | Designation | Source or reference | Identifiers | Additional information |
|---|---|---|---|---|
| Gene (C. elegans) | rnst-2 | | WBGene00019624 | Other name: CELE_K10C9.3 |
| | rpl-5 | | WBGene00004416 | Other name: CELE_F54C9.5 |
| | rpl-25.2 | | WBGene00004439 | Other name: CELE_F52B5.6 |
| | rps-3 | | WBGene00004472 | Other name: CELE_C23G10.3 |
| | rps-0 | | WBGene00004469 | Other name: CELE_B0393.1 |
| Gene (human) | rnaset2 | National Center for Biotechnology Information | Gene ID: 8635 | |
| Strain (C. elegans) | N2 | CGC | RRID:WB-STRAIN:N2 _(ancestral) | wild type (Bristol) |
| | XW1624 | DOI: 10.1126/science.1220281 | | laat-1(qx42) |
| | XW5331 | DOI: 10.1016/j.cell.2010.04.034 | | lgg-1(bp500) |
| | OK286 | DOI: 10.1042/BJ20131085 | RRID:WB-STRAIN:OK286 | pyr-1(cu8) |
| | HZ1690 | DOI 10.1016/j.devcel.2011.06.024 | RRID:WB-STRAIN:HZ1690 | epg-6(bp242) |
| | SP507 | DOI: 10.1042/BJ20131085 | RRID:WB-STRAIN:SP507 | umps-1(mn160) |
| | CF1041 | DOI: 10.1126/science.1139952 | | daf-2(e1370) |
| | HZ1687 | DOI: 10.1083/jcb.201209098 | RRID:WB-STRAIN:HZ1687 | atg-9(bp564) |
| | VC893 | DOI: 10.1016/j.cell.2010.04.034 | RRID:WB-STRAIN:VC893 | atg-18(gk378) |
| | XW10781 | this paper | | rnst-2(qx245) |
| | XW15104 | this paper | | rnst-2(bp555) |
| | HZ1683 | DOI 10.1016/j.devcel.2011.06.024 | RRID:WB-STRAIN:HZ1683 | atg-2(bp576) |
| | CF1903 | CGC | RRID:WB-STRAIN:CF1903 | glp-1(e2144) |
| | XW5097 | RRID:WB-STRAIN:PD4482 | | lmp-1(nr2045) |
| | XW7251 | DOI: 10.1126/science.1220281 | | qxIs354 (P_{ced-1}LAAT-1::GFP) |
| | XW5399 | DOI: 10.1126/science.1220281 | | qxIs257 (P_{ced-1}NUC-1::CHERRY) |
| | XW8056 | DOI: 10.1083/jcb.201602090 | | qxIs430 (P_{scav-3}SCAV-3::GFP) |

*Continued on next page*

*Continued*

| Reagent type (species) or resource | Designation | Source or reference | Identifiers | Additional information |
|---|---|---|---|---|
| | XW10197 | DOI: 10.1083/jcb.201602090 | | *qxIs468* ($P_{myo-3}$LAAT-1::GFP) |
| | XW642 | DOI: 10.1091/mbc.10.12.4311 | | *bIs1* ($P_{vit-2}$VIT-2::GFP) |
| | RT688 | DOI: 10.1126/science.1220281 | | *pwIs281* ($P_{pie-1}$CAV-1::GFP) |
| | XW8293 | DOI: 10.15252/embr.201438618 | | *qxEx4098* ($P_{ced-1}$GFP:ATG-18) |
| | XW12615 | this paper | | *qxIs544* ($P_{rnst-2}$RNST-2::CHERRY) |
| | XW13567 | this paper | | *qxEx6614* ($P_{rnst-2}$RNST-2(H118A)::CHERRY) |
| | XW8783 | this paper | | *qxEx4279* ($P_{myo-3}$LAAT-1::GFP) |
| | XW17778 | this paper | | *qxEx8302* ($P_{rnst-2}$RNASET2-CHERRY) |
| Antibody | anti-RPL-5 (rat polyclonal) | this paper | | WB(1:300) |
| | anti-RPL-25.2 (rat polyclonal) | this paper | | WB(1:300) |
| | anti-RPS-0 (rat polyclonal) | this paper | | WB(1:1000) |
| | anti-RPS-3 (rat polyclonal) | this paper | | WB(1:1000) |
| | anti-CPL-1 (rat polyclonal) | DOI: 10.1126/science.1220281 | | WB(1:500) |
| | anti-LGG-1 (rat polyclonal) | DOI: 10.1016/j.cell.2010.04.034 | | WB(1:1000); Immunostaining(1:1000) |
| | anti-alpha-Tubulin (mouse monoclonal) | Sigma-Aldrich (Missouri, USA) | Cat #T5168; RRID:AB_477579 | WB(1:10000) |
| | anti-SQST-1 (rat polyclonal) | DOI: 10.1016/j.cell.2010.04.034 | | Immunostaining (1:1000) |
| | anti-PGL-3 (rat polyclonal) | DOI: 10.1016/j.cell.2010.04.034 | | Immunostaining (1:1000) |
| | anti-SEPA-1 (rabbit polyclonal) | DOI: 10.1016/j.cell.2010.04.034 | | Immunostaining (1:1000) |
| | anti-HSP-60 (mouse polyclonal) | DOI: 10.1126/science.1220281 | | WB(1:1000) |
| | anti-HEL-1 (rabbit polyclonal) | DOI: 10.1126/science.1220281 | | WB(1:500) |
| | anti-RME-1 (mouse polyclonal) | DOI: 10.1126/science.1220281 | | WB(1:1000) |
| Plasmids for generating transgenic strains | pPD49.26-$P_{rnst-2}$RNST-2::CHERRY | this paper | | Cloning described in 'Plasmid construction' |
| | pPD49.26-$P_{rnst-2}$RNASET2::CHERRY | this paper | | Cloning described in 'Plasmid construction' |
| | pPD49.26-$P_{rnst-2}$RNST-2(H118A)::CHERRY | this paper | | Cloning described in 'Plasmid construction' |
| | pPD49.26-$P_{rnst-2}$RNST-2(cDNA)::CHERRY | this paper | | Cloning described in 'Plasmid construction' |
| Plasmids for protein expression | pET-21b-*rpl-25.2* | this paper | | Cloning described in 'Plasmid construction' |

*Continued on next page*

*Continued*

| Reagent type (species) or resource | Designation | Source or reference | Identifiers | Additional information |
|---|---|---|---|---|
| | pET-21b-*rpl-5* | this paper | | Cloning described in 'Plasmid construction' |
| | pET-21b-*rps-0* | this paper | | Cloning described in 'Plasmid construction' |
| | pET-21b-*rps-3* | this paper | | Cloning described in 'Plasmid construction' |
| Commercial assay or kit | Lysosome Isolation Kit | Sigma-Aldrich (Missouri, USA) | Cat #LYSISO1 | |
| | RNeasy Plus Universal Mini Kit | QIAGEN (Hilden, Germany) | Cat #73404 | |
| Chemical compound, drug | Uridine | Sigma-Aldrich (Missouri, USA) | Cat #U3003 | |
| | Cytidine | Sigma-Aldrich (Missouri, USA) | Cat #C4654 | |
| | DAPI | Vector Laboratories (California, USA) | Cat #H-1200; RRID:AB_2336790 | |
| | LysoTracker red | Invitrogen (Oregon, USA) | Cat #L7528 | |
| Software, algorithm | Velocity | PerkinElmer (Massachusetts, USA) | | |
| | Zen | Carl Zeiss (Oberkochen, Germany) | RRID:SCR_01367 | |

## *C. elegans* strains

*C. elegans* strains were cultured and maintained at 20°C using standard protocols (*Brenner, 1974*). The N2 Bristol strain was used as the wild-type strain except in single nucleotide polymorphism (SNP) mapping, in which Hawaiian strain CB4856 was used. The strains used in this work are listed in the key resources table.

## Isolation, mapping, and cloning of *rnst-2*

The *qx245* and *bp555* mutations were isolated from genetic screens for lysosome- and autophagy-defective mutants, respectively. The *qx245* mutation was mapped to the left side of LG V at the genetic map position −17 (Snp-pkP5076) by single nucleotide polymorphism (SNP) mapping. Transformation rescue experiments were performed and a fosmid clone in this region, *WRM0636aH01*, possessed rescue activity. Whole genome sequencing identified a point mutation in the coding sequence of *K10C9.3* within *WRM0636aH01*. Expression of the long PCR fragment covering the open reading frame of *K10C9.3* fully rescued the *qx245* defect. The sequence of the *K10C9.3* gene was determined in both *qx245* and *bp555* alleles and molecular lesions were identified. *qx245* contained a G-to-A transition that results in a substitution of Gly 119 by Glu, while *bp555* contained a C-to-T transition that generates a premature stop codon after Pro55. Both mutant alleles were backcrossed with N2 strain at least four times before further analyses.

## Lysotracker staining

~50 worms aged to day 3 of adulthood were soaked for 1.5 hr in 80 µl LysoTracker red solution (Invitrogen, Oregon, USA), which was diluted as 1:200 by M9. Worms were then transferred to NGM plates with fresh OP50 and recovered for 2 hr before examination. All steps were performed in the dark.

## Quantification of lysosome number and volume

Fluorescence images of embryos, larvae, and adults expressing LAAT-1::GFP or NUC-1: CHERRY in 10–15 *z*-series (0.5 µm/section) were captured by spinning-disk microscopy. Serial optical sections were reconstituted to 3D view and the number and volume of lysosomes were measured by Velocity software (PerkinElmer, Massachusetts, USA). At least 10 worms were quantified in each strain at each stage.

## Microscopy and imaging analysis

Differential interference contrast and fluorescence images were captured by an inverted confocal microscope (LSM880; Carl Zeiss, Oberkochen, Germany) with 488, 405 and 561 lasers and images were processed and viewed using Zen software (Carl Zeiss, Oberkochen, Germany). All images were taken at 20°C.

## Generation of antibodies

Full-length RPL-5, RPL-25.2, RPS-3 and RPS-0 ribosomal proteins tagged with six Histidine residues (RPL-5-His$_6$, RPL-25.2-His$_6$, RPS-3-His$_6$, RPS-0-His$_6$) were expressed in *E. coli*. The recombinant proteins were purified by Ni-NTA agarose beads and the eluted RPL-5-His$_6$, RPS-3-His$_6$ and RPS-0-His$_6$ proteins were used to raise polyclonal antibodies in rat. For RPL-25.2-His$_6$, a specific protein band was excised from the SDS-PAGE and used to generate polyclonal antibody in rat. The RPL-5 and RPS-0 polyclonal antibodies were further purified using RPL-5 or RPS-0 recombinant proteins. All polyclonal antibodies were generated in the Antibody Center at NIBS (National Institute of Biological Sciences, Beijing, China).

## Western blot analysis and immunostaining

Mix-staged embryos of *C. elegans* were collected from aged adults by bleaching. Embryonic and worm lysates (day three adults) were obtained and analyzed by western blot using anti-LGG-1 antibodies and anti-α-tubulin antibodies (Sigma-Aldrich, Missouri, USA).

For immunostaining, mix-staged embryos were fixed by ice-cold methanol and acetone on glass sheets treated by 1% poly-lysine. After blocking with 1% BSA and 10% fetal calf serum in phosphate buffered saline (PBS), the samples were incubated with anti-LGG-1(1:1000 dilution), anti-SQST-1 (1:1000 dilution), anti-SEPA-1 (1:1000 dilution) or anti-PGL-3 (1:1000 dilution) antibodies at 4°C overnight. The samples were washed three times in PBST (PBS + 0.2% Tween 20) and incubated with the secondary antibody (1:200 dilution) (Jackson ImmunoResearch, Pennsylvania, USA) for 80 min at room temperature in the dark. The samples were washed three times, then stained with DAPI (Vector Laboratories, California, USA) and visualized using a Zeiss LSM880 confocal microscope.

## Lysosome purification and examination of RNA accumulation

Lysosomes were purified from adult worms by a Lysosome Isolation kit (LYSISO1; Sigma-Aldrich, Missouri, USA) as described previously with modifications (*Liu et al., 2012*). In brief, synchronized larvae one worms were cultured at 20°C to day 3 of adulthood and collected. Worm pellets were suspended in the extraction buffer supplied in the Lysosome Isolation kit and ruptured by glass beads with the FastPrep−24 Instrument (MP Biomedicals, Ohio, USA). The worm lysate was centrifuged at 14,000 rpm to precipitate the crude lysosomes. The crude lysosomes were diluted in 19% Optiprep Density Gradient Medium Solution and further separated on a sucrose density gradient (from bottom to top: 27%, 19%, 16%, 8%).

After centrifugation, lysosome enrichment was determined by the amount of processed CPL-1 (lysosomal cathepsin) in different fractions that contained the same amount of total proteins (from bottom to top: B1-4). The purity of the lysosome fraction (B3) was examined by antibodies that detect different intracellular organelles.

The purified lysosome fractions from different strains were examined for accumulation of ribosomal proteins by western blot and ribosomal RNAs by RNA purification (73404; QIAGEN, Hilden, Germany). The purified RNAs were separated and examined by agarose gel electrophoresis. At least three independent experiments were performed in each strain and average data are shown in *Figure 3* and *Figure 3—figure supplement 1*.

## Examination of embryonic development, larval development and lifespan

To examine embryonic development, ~20 young adult worms (24 hr post L4) were placed on a NGM plate with OP50 for 2 hr. The worms were then removed and the eggs laid on the plate were followed. After 10 hr, newly hatched L1 worms were counted and transferred every hour until no new L1 worms hatched out. The percentage of eggs that hatched every hour was calculated to show the progress of embryonic development, and the total percentage of hatched L1 was quantified to

determine level of embryonic lethality. More than 150 embryos were quantified in each strain. At least three independent experiments were performed in each strain and the average result from three experiments is shown in *Figure 5*, *Figure 6*, *Figure 6—figure supplements 1* and *2*. To analyze larval development, 100 L1 worms were collected. After 48 hr incubation, newly developed L4 worms were counted and transferred every hour until no live larvae were present on the plate. The total percentage of fully developed L4 was quantified to determine the level of larval arrest. At least three independent experiments were performed in each strain and the average result from three experiments is shown in *Figure 4—figure supplement 1C* and *Figure 6—figure supplement 1E*. Lifespan assays were performed at 20°C or 25°C [*glp-1(e2144)* and the control strains] as described previously (*Hansen et al., 2005*). About 150 L4 worms (day 0) were picked to NGM plates with fresh OP50, 15 worms per plate. The surviving worms were counted every 2 days and were transferred to new plates to avoid interference from the progeny. The worms that crawled off the plate, exploded, bagged, or became contaminated were discarded. At least three independent experiments were performed for each strain. The representative survival curve is shown in *Figure 5D and F*, *Figure 6—figure supplement 1F and H* and the mean lifespan from three experiments is shown in *Figure 5E and G*, *Figure 6—figure supplement 1G and I*.

## Quantification of L1 survival under starvation treatment

Mix-staged embryos were collected and placed in M9 buffer, so that their growth could be synchronized to the larvae 1 (L1) stage. The L1 worms were transferred equally to 96-well plates, with about 200 larvae per well. The worms were transferred to NGM plates with fresh OP50 every 2 days and the surviving larvae were quantified. At least three independent experiments were performed for each strain. The representative survival curve is shown in *Figure 4U* (left panel) and the mean lifespan from three independent experiments is shown in *Figure 4U* (right panel).

## RNAi

The bacteria-feeding RNAi protocol was used as described before (*Kamath and Ahringer, 2003*). In *pyr-1* RNAi (D2085.1; 3117–4214 nt) and *umps-1* RNAi (T07C4.1; 274–1400 nt) experiments,~30 L1 or L2 worms (P0) were placed on the RNAi plates. Embryos of the F1 generation were collected and quantified for survival throughout embryonic development.

## Nucleotide supplementation

Uridine and cytidine (Sigma-Aldrich, Missouri, USA) were dissolved in distilled water to a final concentration of 1M and this stock solution was stored at −20°C. The stock solution of uridine or cytidine was supplied to both NGM plates and OP50 culture spotted onto NGM plates as a 1:10 dilution. L4 larvae (P0) were placed on uridine- or cytidine-containing plates and cultured to the next generation (F1). The survival of the F2 embryos was quantified.

## Statistical analysis

The standard deviation (SD) was used as y-axis error bars for bar charts plotted from the mean value of the data. Data derived from different genetic backgrounds were compared by Student's two-tailed unpaired t test, one-way analysis of variance (ANOVA) with Tukey's *post hoc* test or two-way ANOVA with the Bonferroni *post hoc* test. Data were considered statistically different when $p < 0.05$. $p < 0.05$ is indicated with single asterisks, $p < 0.001$ with double asterisks and $p < 0.0001$ with triple asterisks (0.001 in a two-way ANOVA analysis).

## Plasmid construction

To construct $P_{rnst-2}$RNST-2::CHERRY, $P_{rnst-2}$RNST-2 was amplified from N2 genomic DNA using primers PDFZ482/PYBL29 and was ligated to pPD49.26-nCHERRY3 through the Nhe I/Nco I sites. To generate the $P_{rnst-2}$RNASET2::CHERRY reporter, the 2.1 kb promoter of *rnst-2* was amplified from N2 genomic DNA using primers PYBL42/PYBL43 and cloned into pPD49.26-nCHERRY3 through the BamH I site. RNASET2 of *Homo sapiens* was amplified from a human cDNA library using the primers PDFZ487/PDFZ488 and cloned into $P_{rnst-2}$nCHERRY3 through the Nhe I/Nco I sites. $P_{rnst-2}$RNST-2 (H118A)::CHERRY was generated through PCR-based site-directed mutagenesis using the primers PDFZ485/PDFZ486 on $P_{rnst-2}$RNST-2(cDNA)::CHERRY. The cDNA of *rnst-2* was amplified from a *C.*

*elegans* cDNA library (Invitrogen, Oregon, USA) using primers PYBL32/PYBL41 and cloned into P$_{rnst-2}$nCHERRY3 through the Nhe I site. The RNST-2(H118A) fragment was digested from P$_{rnst-2}$RNST-2 (H118A)::CHERRY after mutagenesis, and re-ligated to P$_{rnst-2}$nCHERRY3 through the Nhe I site. For the protein expression vectors, the cDNAs of *rpl-25.2*, *rps-3* and *rps-0* were amplified from a *C. elegans* cDNA library (Invitrogen, Oregon, USA) using primers PYBL64/PYBL65, PYBL70/PYBL71 and PYBL67/PYBL68, respectively, and ligated to pET21b vector through the Nde I/Xho I sites. The cDNA of *rpl-5* was amplified from a *C. elegans* cDNA library (Invitrogen, Oregon, USA) using the primers of PYBL99/PYBL101 and ligated to pET21b vector through the Nhe I/Xho I sites.

## Primers used for plasmid construction

| Primer | Sequence (5' to 3') |
|---|---|
| PDFZ482 | GCCCATGGCATTTCCAATATTTTTGATAGCGCTCC |
| PYBL29 | GCGCTAGCCCGTGGGAGTAATGTTGC |
| PYBL42 | GCGGATCCTTCTGGAAATCTTGCGTGAT |
| PYBL43 | GCGGATCCGGCGACTACTGTAAACGA |
| PDFZ487 | CGGCTAGCATGCGCCCTGCAGCCCTGCGCGG |
| PDFZ488 | GCCCATGGCATGCTTGGTCTTTTTAGGTGGGG |
| PDFZ485 | GAAACACGAGTATGATAAGGCCGGGACATGTGCTCAAAG |
| PDFZ486 | CTTTGAGCACATGTCCCGGCCTTATCATACTCGTGTTTC |
| PYBL32 | GCGCTAGCATGAAACTTCTCCTTCTTCTCT |
| PYBL41 | GCGCTAGCATTTCCAATATTTTTGATAGCG |
| PYBL64 | GCCATATGATGGCTCCGTCATCAAACAAAG |
| PYBL65 | GCCTCGAGGATGAATCCGATCTTGTTGGCA |
| PYBL70 | GCCATATGATGGCTGCCAATCAAAACGTGA |
| PYBL71 | GCCTCGAGTTGAACTGGAGCAACTGGTTGA |
| PYBL67 | GCCATATGATGTCAGGCGGTGCCGCTCATT |
| PYBL68 | GCCTCGAGCCAGTTAGACTGGGTTGGAGCG |
| PYBL99 | GCGCTAGCATGGGTCTCGTCAAGGTCATTA |
| PYBL101 | GCCTCGAGCTCCTGTTGCTCCTTGAGTT |

## Acknowledgements

We thank Dr. Mengqiu Dong (National Institute of Biological Sciences) for antibodies and Dr. Isabel Hanson for editing services. Some strains were provided by the CGC, which is funded by NIH Office of Research Infrastructure Programs (P40OD010440). This work was supported by the Ministry of Science and Technology (2016YFA0500203), the National Science Foundation of China (3163001, 91754203) and the Strategic Priority Research Program of the Chinese Academy of Sciences (XDB19000000) to X W. The authors declare no competing financial interests.

## Additional information

### Funding

| Funder | Author |
|---|---|
| Ministry of Science and Technology of the People's Republic of China | Xiaochen Wang |
| National Natural Science Foundation of China | Xiaochen Wang |
| Chinese Academy of Sciences | Xiaochen Wang |

The funders had no role in study design, data collection and interpretation, or the decision to submit the work for publication.

### Author contributions

Yubing Liu, Conceptualization, Formal analysis, Investigation, Visualization, Writing—original draft; Wei Zou, Li Wang, Yan Ma, Investigation, Visualization; Peiguo Yang, Investigation; Hong Zhang,

Supervision, Writing—review and editing; Xiaochen Wang, Conceptualization, Supervision, Funding acquisition, Writing—original draft, Project administration

### Author ORCIDs
Xiaochen Wang ⓘD http://orcid.org/0000-0002-4344-0925

### Decision letter and Author response
Decision letter https://doi.org/10.7554/eLife.36588.029
Author response https://doi.org/10.7554/eLife.36588.030

## Additional files

### Supplementary files
• Transparent reporting form
DOI: https://doi.org/10.7554/eLife.36588.026

### Data availability
All data generated or analyses during this study are included in the manuscript and supporting files.

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
