## [Decision Letter]

Thank you for submitting your article "Autophagy-dependent rRNA degradation is essential for maintaining nucleotide homeostasis during *C. elegans* development" for consideration by *eLife*. Your article has been reviewed by three peer reviewers, one of whom is a member of our Board of Reviewing Editors, and the evaluation has been overseen by a Reviewing Editor and Ivan Dikic as the Senior Editor. The reviewers have opted to remain anonymous.

The reviewers have discussed the reviews with one another and the Reviewing Editor has drafted this decision to help you prepare a revised submission.

Summary:

This study reports that *C. elegans* RNST-2, a lysosomal T2 family endoribonuclease, is required for autophagic degradation of ribosomal RNAs and is critical for development and survival of *C. elegans* embryos and larvae. Importantly, the authors find that simultaneous suppression of RNST-2 and de novo synthesis of pyrimidine nucleotides causes complete lethality and that this phenotype can be rescued by uridine/cytidine supplementation. These results suggest that RNST-2-dependent lysosomal RNA turnover is critical for a pyrimidine nucleotide recycling, general lysosomal function, and worm development. This work potentially represents a significant advancement in the field. However, there are several points that should be clarified.

Essential revisions:

(Comments 1, 2, and 3 are related)

1) One of the main conclusions is that lysosomal turnover of RNAs is important for providing nucleotides. Although this can be hypothesized by the data, the levels of nucleotides are not directly measured in mutants. Quantification of nucleotides in wild-type and *rnst-2, pyr-1, pyr-1;rnst-2*, and some *atg* mutants would directly support this conclusion.

2) In Figure 4F and G, although uridine or cytidine supplementation rescues embryonic lethality of *rnst-2* mutants, it does not improve embryonic viability of *rnst-2* mutants. It could indicate that reduced embryonic lethality is not caused primarily by defects in the supply of pyrimidines. Instead, other functions of RNST-2 could be the cause of embryonic lethality of *rnst-2* mutants. For example, a general defect in autophagy could be the reason behind both embryonic lethality and synthetic lethality phenotypes. This point should be clarified to prove the recycling hypothesis.

3) As autophagy mutants such as *lgg-1(bp500), atg-18(gk378), atg-2(bp576)* also show partial embryonic lethality, it is still unclear whether this lethality is caused by a defect in RNA degradation or other molecules. The embryonic lethal phenotype of autophagy mutants should also be tested with uridine or cytidine supplementation.

4) Another important conclusion is that general lysosomal function is impaired in *rnst-2* mutants. If this is due to accumulation of autophagy-derived RNAs, is not only the size of lysosomes (Figure 3E) but also their function (e.g. degradation of endocytosed cargos) restored by additional deletion of autophagy genes such as *atg-2*?

5) The data presented here also suggest that the autophagy pathway itself is affected in *rnst-2* mutants. In Figure 4—figure supplement 1, the accumulation of LGG-1-I is difficult to explain simply by lysosomal dysfunction. Generally, impairment of lysosomal function causes accumulation of LGG-1/ATG8-II, but not -I. The result rather suggests that autophagy induction or autophagosome formation may also be affected. To further characterize the autophagy pathway, additional assays (i.e. measurement of autophagic flux using bafilomycin A1 and testing accumulation of phagophore markers such as ATG1 and ATG18). This is important because in other organisms (e.g., *Arabidopsis*) autophagosomes accumulate in *rnst-2* mutants in an attempt to compensate for inhibition of RNA degradation.

[Editors' note: further revisions were requested prior to acceptance, as described below.]

Thank you for resubmitting your work entitled "Autophagy-dependent rRNA degradation is essential for maintaining nucleotide homeostasis during *C. elegans* development" for further consideration at *eLife*. Your revised article has been favorably evaluated by Ivan Dikic (Senior Editor), a Reviewing Editor, and two reviewers.

The manuscript has been improved but there are some remaining issues that need to be addressed before acceptance, as outlined below:

We appreciate that you have made a substantial effort to address the comments and criticisms of the first review round. We have assessed this revised version in consultation with the two original reviewers.

1) We also think that the new results shown in Author response image 1 is indeed paradoxical; the level of uridine in *rnst-2;pyr-1* mutant embryos is higher than in control embryos. This is contradictory to the main result that uridine supplementation can suppress the lethality of the double mutants. However, we understand that it may be difficult to accurately measure the nucleotide levels using heterogeneous tissues in unhealthy embryos (with the intact purine synthesis pathway) and the results may not reflect the situation in key organs causing the phenotype. We agree that the other data support your recycling hypothesis. Given this situation, it would be fair to state that you have tried to measure the nucleoside levels and observed a reduction in cytidine but not in guanosine (even without showing the data) and discuss potential limitations or difficulties of this experiment.

2) Although it would be difficult to clearly separate phenotypes caused by a defect in general autophagy from those by a defect in nucleoside supply, this point should be more clearly discussed in the Discussion section. The phenotype should be caused by both defects.

3) During the revision, a related paper on autophagy-mediated nucleotide recycling was published by David Sabatini's group (Wyant et al., 2018). You may want to cite this paper and discuss a relationship with this paper.

---

## [Author Response]

Essential revisions:(Comments 1, 2, and 3 are related)1) One of the main conclusions is that lysosomal turnover of RNAs is important for providing nucleotides. Although this can be hypothesized by the data, the levels of nucleotides are not directly measured in mutants. Quantification of nucleotides in wild-type and rnst-2, pyr-1, pyr-1;rnst-2, and some atg mutants would directly support this conclusion.

As suggested by the reviewer, we measured the levels of 4 nucleosides (adenosine, guanosine, cytidine and uridine) and UMP by LC/MS (Agilent 1290 HPLC coupled to an Agilent 6495 Triple Quadrupole mass spectrometer). We first measured the nucleoside levels in samples prepared from mix-staged worms, but they gave very big variations, likely due to asynchronization of the worms. We then switched to measurements using mix-staged embryos, which are hard to collect but showed less variations than the mix-staged worm samples. We utilized RNAi treatment in these experiments because (1) *pyr-1;rnst-2* double mutants do not produce viable progeny and we were unable to collect enough embryos for measurements; (2) *rnst-2;pyr-1(RNAi)* exhibited synthetic embryonic lethality as in double mutants, but the RNAi treatment can be achieved on a sufficiently large scale to obtain enough embryos for LC-MS analyses. We found that levels of adenosine, guanosine, cytidine, uridine and UMP were all obviously decreased in *rnst-2(qx245)* and *pyr-1(RNAi)* embryos compared to wild type, consistent with the role of RNST-2 and PYR-1 in maintaining nucleotide homeostasis (Author response image 1). In *rnst-2;pyr-1(RNAi)*, the cytidine level appeared to be further reduced, while levels of adenosine and UMP were similar to those in single mutants (Author response image 1). Unexpectedly, however, levels of guanosine and uridine were higher in *rnst-2;pyr-1(RNAi)* than in wild type and single mutants (Author response image 1). We do not understand why guanosine and uridine levels increased in *rnst-2;pyr-1(RNAi)*. Since RNA inactivation reduces but does not completely disrupt PYR-1 function, it is possible that feed-back responses may be elicited in *rnst-2;pyr-1(RNAi)*, which lead to increased guanosine and uridine through de novo or salvage pathways. However, this is pure speculation and we do not have a straight forward approach to test this hypothesis due to the high complexity and our incomplete understanding of the pyrimidine and purine metabolism pathways in worms. We therefore did not include these data in the revised manuscript. Nevertheless, except for the increase in guanosine and uridine levels in *rnst-2;pyr-1(RNAi)* embryos, the nucleoside measurement results are all consistent with the genetic and cell biology data and support the idea that lysosomal turnover of RNA is important for providing nucleotides.

**Author response image 1. respfig1:** Quantitative analysis of nucleosides. Mix-staged embryos were collected and ground. The metabolites were extracted, and nucleosides and UMP were analyzed by LC/MS. The level of each nucleoside is presented as the absolute concentration, and the UMP level is presented as normalized intensity on the basis of the total peak area.

2) In Figure 4F and G, although uridine or cytidine supplementation rescues embryonic lethality of rnst-2 mutants, it does not improve embryonic viability of rnst-2 mutants. It could indicate that reduced embryonic lethality is not caused primarily by defects in the supply of pyrimidines. Instead, other functions of RNST-2 could be the cause of embryonic lethality of rnst-2 mutants. For example, a general defect in autophagy could be the reason behind both embryonic lethality and synthetic lethality phenotypes. This point should be clarified to prove the recycling hypothesis.3) As autophagy mutants such as lgg-1(bp500), atg-18(gk378), atg-2(bp576) also show partial embryonic lethality, it is still unclear whether this lethality is caused by a defect in RNA degradation or other molecules. The embryonic lethal phenotype of autophagy mutants should also be tested with uridine or cytidine supplementation.

As suggested by the reviewer, we examined embryonic lethality in *lgg-1, atg-18* and *atg-2* without and with uridine or cytidine supplements. Like in *rnst-2* single mutants, uridine and cytidine supplements did not rescue the partial embryonic lethality in these autophagy-defective mutants, although cytidine supplement increased the viability of *atg-18* embryos (Figure 6—figure supplement 2C, D). These data and our finding that loss of *rnst-2* partially impairs autophagy together suggest that a general autophagy defect may contribute to the partial embryonic lethality in *rnst-2* and autophagy-defective mutants.

We found that in *pyr-1* and *umps-1* mutants, which have a blockage in de novo synthesis of pyrimidines, loss of RNST-2, but not LAAT-1 (lysosomal amino acid transporter) or LMP-1/LAMP1, leads to complete embryonic lethality. This indicates that loss of RNST-2 function, but not a general defect of lysosomes, leads to the synthetic embryonic lethality (Figure 6A, B and Figure 6—figure supplement 1C-E). Although supplements of uridine or cytidine did not obviously improve the viability of *rnst-2* single mutants, these treatments efficiently suppressed the synthetic lethality in *pyr-1;rnst-2* double mutants (Figure 6C, E). This strongly suggests that the synthetic lethality is mainly caused by defective supply of nucleotides.

We have included the new rescue data in the revised manuscript to indicate that a general autophagy defect contributes to the embryonic lethality in *rnst-2* and autophagy-defective mutants.

4) Another important conclusion is that general lysosomal function is impaired in rnst-2 mutants. If this is due to accumulation of autophagy-derived RNAs, is not only the size of lysosomes (Figure 3E) but also their function (e.g. degradation of endocytosed cargos) restored by additional deletion of autophagy genes such as atg-2?

In *rnst-2*, we examineddegradation of the endocytic cargos CAV-1 and VIT-2, and of apoptotic cells, which are engulfed and degraded in lysosomes through the phagocytic pathway. We found that CAV-1, VIT-2 and apoptotic cell corpses are properly degraded in *rnst-2* as in wild type (Figure 3—figure supplement 2). Thus, loss of RNST-2 affects degradation of autophagic but not endocytic or phagocytic cargos.

5) The data presented here also suggest that the autophagy pathway itself is affected in rnst-2 mutants. In Figure 4—figure supplement 1, the accumulation of LGG-1-I is difficult to explain simply by lysosomal dysfunction. Generally, impairment of lysosomal function causes accumulation of LGG-1/ATG8-II, but not -I. The result rather suggests that autophagy induction or autophagosome formation may also be affected. To further characterize the autophagy pathway, additional assays (i.e. measurement of autophagic flux using bafilomycin A1 and testing accumulation of phagophore markers such as ATG1 and ATG18). This is important because in other organisms (e.g., Arabidopsis) autophagosomes accumulate in rnst-2 mutants in an attempt to compensate for inhibition of RNA degradation.

As suggested by the reviewer, we performed additional experiments to further characterize the autophagy process in *rnst-2* mutants. Unfortunately, we do not have an efficient way to completely block lysosome function in *C. elegans* (bafilomycin A1 treatment did not work well), and therefore we are unable to measure autophagic flux precisely in *rnst-2* mutants. To further examine the autophagy process in *rnst-2*, we did the following experiments:

1) We examined accumulation of PGL granules, which are efficiently removed by selective autophagy during embryogenesis. This process is very sensitive to changes in autophagy activity (Zhang et al., 2009; Tian et al., 2010). The PGL granule component PGL-3 and the bridging molecule SEPA-1 failed to be removed and thus persisted in *atg-2* mutants (Figure 4—figure supplement 1). However, PGL-3 and SEPA-1 were removed in *rnst-2* mutants as in wild type, indicating that autophagic clearance of PGL granules is normal (Figure 4—figure supplement 1).

2) SQST-1/p62 associates with various protein aggregates that are removed by autophagy, while LGG-1/ATG8 associates with autophagic structures from early to late stages. Both SQST-1 and LGG-1 are substrates of autophagy. We found that SQST-1 and LGG-1 puncta were removed in wild type but were present in late-staged embryos in *rnst-2* mutants (Figure 4 and Figure 4—figure supplement 1). In *atg-2* mutants, however, SQST-1 and LGG-1 puncta persisted in embryos at both early and late stages (Figure 4). These data suggest that the autophagy process is partially impaired or delayed in *rnst-2(lf)*, causing accumulation of SQST-1 and LGG-1 in late-staged embryos.

3) *rnst-2* mutants accumulated both LGG-1-I and LGG-1-II (lipid-conjugated form of LGG-1) at both embryonic and adult stages, consistent with partial impairment of autophagy (Figure 4).

4) We found that GFP::ATG-18, which associates with early autophagic structures, was diffuse in the cytoplasm in both wild type and *rnst-2* mutant embryos, suggesting that autophagosome formation is not blocked (Figure 4—figure supplement 2).

5) Loss of *rnst-2* leads to shortened lifespan of L1 larvae in the absence of food, a process that requires autophagy activity.

Altogether, these data suggest that the autophagy process is partially impaired in *rnst-2* mutants. We suspect that the gradual buildup of undigested rRNA and ribosomal proteins in *rnst-2* lysosomes leads to impaired autophagy kinetics and thus causes a reduction in autophagic flux. This is in fact consistent with the notion that dysregulation of autophagy serves as a common mechanism underlying lysosomal storage diseases that are characterized by the accumulation of undegraded metabolites in lysosomes.

[Editors' note: further revisions were requested prior to acceptance, as described below.]

The manuscript has been improved but there are some remaining issues that need to be addressed before acceptance, as outlined below:We appreciate that you have made a substantial effort to address the comments and criticisms of the first review round. We have assessed this revised version with consultation with the two original reviewers.1) We also think that the new results shown in Author response image 1 in the rebuttal letter is indeed paradoxical; the level of uridine in rnst-2;pyr-1 mutant embryos is higher than in control embryos. This is contradictory to the main result that uridine supplementation can suppress the lethality of the double mutants. However, we understand that it may be difficult to accurately measure the nucleotide levels using heterogeneous tissues in unhealthy embryos (with the intact purine synthesis pathway) and the results may not reflect the situation in key organs causing the phenotype. We agree that the other data support your recycling hypothesis. Given this situation, it would be fair to state that you have tried to measure the nucleoside levels and observed a reduction in cytidine but not in guanosine (even without showing the data) and discuss potential limitations or difficulties of this experiment.

We have included the statement regarding nucleoside measurement in the Discussion section as suggested, and discussed the limitation of the experiments.

2) Although it would be difficult to clearly separate phenotypes caused by a defect in general autophagy from those by a defect in nucleoside supply, this point should be more clearly discussed in the Discussion section. The phenotype should be caused by both defects.

We have revised the Discussion section as suggested to indicate clearly that the development phenotype may be caused by defects in both general autophagy and nucleotide supply.

3) During the revision, a related paper on autophagy-mediated nucleotide recycling was published by David Sabatini's group (Wyant et al., 2018). You may want to cite this paper and discuss a relationship with this paper.

We have cited the paper published by David Sabatini’s group and discussed the potential relationship with our study (subsection “Ribosomal RNA degradation through the autophagy-lysosome pathway is important in maintaining nucleotide homeostasis essential for animal development”).